# Prognostic Survival Significance of Signet Ring Cell (SRC) Gastric Cancer: Retrospective Analysis from a Single Western Center

**DOI:** 10.3390/jpm13071157

**Published:** 2023-07-19

**Authors:** Luigina Graziosi, Elisabetta Marino, Nicola Natalizi, Annibale Donini

**Affiliations:** General and Emergency Surgery of Santa Maria della Misericordia Hospital, University of Perugia, 06132 Perugia, Italy; elisabetta.marino1986@gmail.com (E.M.); nicolanatalizi@gmail.com (N.N.); annibale.donini@unipg.it (A.D.)

**Keywords:** gastric cancer, signet ring cell, poorly cohesive carcinoma

## Abstract

Introduction: Signet ring cell carcinoma accounts for 35% to 45% of all gastric cancer. Despite the acknowledgment of its more aggressive pathological features, various controversies surrounding this topic still exist. Thus, we investigate the clinical pathological characteristics and survival prognostic significance of signet ring cell components in patients affected by gastric cancer. Methods: From January 2004 to December 2020, in a retrospective study, we enrolled 404 patients with gastric cancer who were curatively treated in our department. The male-to-female ratio was 249/142, and the median age was 75 (range 37–94). We dichotomized patients into two groups (75 patients vs. 316 patients) based on the signet ring cell presence; according to preoperative, operative, and postoperative characteristics, we performed a univariate and multivariate analysis for overall survival. Results: Signet ring cell carcinoma indicated an increasing incidence trend over the time analyzed. Overall median survival of signet ring cell and non-signet ring cell carcinoma were, respectively, 16 vs. 35 months, *p* < 0.05. In early gastric cancer, the prognosis of the signet ring cell is better than that of the non-signet ring cell, as opposed to advanced cancer. Among the entire population in the multivariate analysis, the only independent factors were preoperative serum albumin level, complete surgical resection, level of lymphadenectomy, and pathological stage. Recurrence occurred more frequently in patients affected by signet ring cell, but in our data, we could not identify a peculiar site of recurrence. Conclusions: Signet ring cell carcinoma has a specific oncogenetic phenotype and treatment resistance heterogeneity; however, it is not always associated with poor prognosis. According to our results, a radical surgical procedure associated with an adequate lymphadenectomy should be advocated to improve patients survival. Gastric cancer patients with signet ring cell components should draw clinicians’ attention.

## 1. Introduction

Gastric cancer (GC) is the fifth most common cancer and the third most common cause of death related to cancer. The incidence of GC is different around the world. In fact, the incidence is higher in East Asian countries (i.e., China and Japan), in East European countries, and in other European countries, such as Italy. This distribution is associated with several causes, such as Helicobacter pylori (HP) infection, lifestyle, diet (for example, a low consumption of vegetables and fruits and obesity), genetic family history of gastric cancer, and the use of tobacco [1]. 

In gastric cancer (GC) patients, adenocarcinoma is the most common pathological tumor type, accounting for approximately 90% of all gastric cancer cases. 

Gastric signet ring cell carcinoma (GSRCC) is a subtype of gastric adenocarcinoma that is related to aggressive malignancy behavior and poor prognosis. 

Literature reports that the incidence of GSRCC is constantly increasing in Asia, the United States, and Europe, accounting for 35% to 45% of gastric adenocarcinoma cases in recent studies [2,3].

GSRCC is a well-histopathologically-described subtype; however, since 2019, with the 5th World Health Organization (WHO) classification of tumors of the digestive system [4,5], GSRCC has been recognized as a particular and distinguished category belonging to the poorly cohesive adenocarcinoma type. 

Among GSRCC, we can identify two subtypes that depend on the percentage of cells demonstrating poorly peculiar characteristics, ending up with GSRCC type A, in which over 90% of cells show those features, and GSRCC not otherwise specified (NOS), in which less than 90% of the cells can be classified as above [4]. 

Epidemiological changes are occurring within Western countries, indicating a decrease in the intestinal subtype in favor of an increase in the diffuse type, particularly in GSRCC subtypes, especially in younger patients [6,7]. GSRCC of the stomach and GEJ have distinct characteristics such as younger age at presentation, female predominance, advanced stage, lymphatic spread, peritoneal metastasis, rapid progression, and finally a worse prognosis [3].

Although there are many studies on the prognostic value of GSRCC, they are strongly inconsistent in their results, resulting in no meaningful conclusions. 

Major controversies are about gastric signet ring cell carcinoma’s correlation to disease stage and its chemo-sensitivity; the role of neo-adjuvant treatment in GSRCC patients and also an up-front surgical approach. To date, there is still no clear mechanism conferring chemoresistance to GSRCC; the characteristics of SRC, such as the intra-cytoplasmic vacuole of mucinous content, could be the reason why there is less response to chemotherapy [8,9].

Moreover, GSRCC’s prognosis is worse because it has different intrinsic molecular characteristics compared to gastric adenocarcinoma.

Fourgeaud et al. described GSRCC as a separate entity from gastric cancer. In particular, they studied the expression of heparanase (HPA) in gastric cancer tissues. In fact, HPA is an enzyme that has a lot of functions, such as up-regulating the vascular endothelial growth factors (VEGF) A and C and activating a pathway inside the cell responsible for the survival, migration, and proliferation of tumor cells. They found that the overexpression of HPA in GSRCC is associated with a worse prognosis than in gastric adenocarcinoma [10]. 

Controversies still remain as some studies evaluated show that GSSRCC has a better prognosis, while others conclude that SRC has a worse prognosis or does not harbor any different prognostic features compared to nGSSRCC. Another important point is that most information is derived from eastern countries; few data are coming from the west or from our country.

For this reason, we would like to draw a picture of our GSRCC population by retrospectively investigating its clinical and pathological characteristics and subsequently correlating them to overall survival.

## 2. Methods

From January 2004 to December 2020, 404 patients affected by gastric cancer were curatively and surgically treated in our surgical and oncological department. 

The ethical committee of Umbria (CEAS) granted permission to collect patients’ data, permit FI00001, n. 2266/2014, granted on 19 February 2014, and by the University of Perugia Bioethics Committee, permit FIO0003, n. 36348, granted on 6 May 2020. Informed written consent was obtained by each patient before surgery.

All procedures followed were under the ethical standards of the responsible committee on human experimentation (institutional and national) and the Helsinki Declaration of 1964 and later versions.

After excluding patients with incomplete data, we enrolled 391 patients in the analysis. The male-to-female ratio was 249/142; males represented 63.2% and females 46.8% of the entire population. The median age was 75 years old, with a range of 37–94. 

No hereditary diffuse gastric cancer (HDGC) patients belonged to this series. 

We dichotomized patients into two groups (75 patients vs. 316 patients) based on the gastric signet ring cell (GSRC) presence and performed a univariate analysis for overall survival, according to preoperative, operative, and postoperative GC patients’ features.

Among the preoperative features, we evaluated the following variables:Age;Gender;Serum albumin level;Inflammatory markers such as neutrophils-to-lymphocytes ratio (NLR)Tumor location.

Among the surgical-related features, we evaluated the following variables:Type of surgery;Number of harvested nodes;Lymph node ratio.

Among the post-operative features, we evaluated the following variables:Early gastric cancer vs. advanced gastric cancer;pT;pN;Peritoneal cytology status;Veno-lymphatic infiltration;Perineural infiltration;*p* Stage;Completeness of surgical procedure;Recurrences sites;Neo/and Adjuvant chemotherapy.

Venous blood samples were taken either the day before surgery or a few days immediately before and collected in an ethylenediaminetetraacetate acid-containing tube, according to other studies present in the literature. 

The normal range of the white blood cell (WBC) count was between 4000 and 10,800 cells/mm^3^.

The normal range of albumin was between 3.5 g/dL and 5.2 g/dL. 

We calculated baseline NLR as a neutrophil count divided by lymphocyte count.

We evaluated the lymph node ratio as the ratio between positive lymph nodes and the total number of lymph nodes retrieved. 

We dichotomized patients at the median value of NLR, lymph node ratio, harvested Lymph nodes, and serum albumin level (3.5 g/dL). 

Pathological stage and parameters were evaluated according to the AJCC/TNM 8th edition [11]. We also performed a multivariate analysis.

All patients were followed up regularly until death: every 6 months for the first 2 years after surgery and every year thereafter for at least 10 years.

### Statistical Analysis

Patients’ descriptive analysis was generated, and we investigated their differences using Student’s *t*-test for quantitative data; the normality test accorded to D’Agostino-Pearson was performed, and when not passed, quantitative data were compared using the Mann-Withney test. For qualitative data, we used either Fisher’s exact test or the Chi-square test. 

Overall survival (OS) analysis was carried out with the method of Kaplan-Meier, and differences were evaluated using the log-rank test. We subsequently evaluated only variables that achieved statistical significance in the univariate analysis in the multivariate analysis using Cox’s proportional hazard regression model. A *p*-value of less than 0.05 was considered statistically significant.

All statistical analyses were performed using MedCalc Statistical Software version 14.8.1 (MedCalc Software, Ostend, Belgium), PRISM 7.2 Graph PAD, and SPSS, IBM version 23. 

## 3. Results

We evaluated the clinical pathological feature distribution among the two groups, as shown in Table 1.

The overall survival of our entire population was analyzed, with the analysis demonstrating a median survival of 28 months and a 5Y-OS of 40%.

As also known from the epidemiological changes around the world our series, demonstrated an increasing incidence trend over the time analyzed when compared to the N-SRCC population (*p* = 0.05); Figure 1.

The overall median survival lengths of GSRCC and N- GSRCC were, respectively, 16 vs. 35 months. A statistically significant difference, however, is not reached; *p* > 0.05 (Figure 2).

As already stated, our population’s median age was 75 years old; subdividing the series according to the data, we detected that patients with GSRCC adenocarcinoma were mostly below the median age value (*p* < 0.05). 

This has a statistically significant impact on survival in both groups, as we see in Figure 3, with patients above 75 years old indicating a worse prognosis (*p* < 0.05).

Males and females were statistically different distributed among the N-GSRCC group, showing an impact on the prognosis as overall survival was statistically higher in female patients (Figure 4a,b); whereas within the GSRCC patients’, gender was not a significant factor for survival.

Among the pathological-related factors, we determined that perineural invasion was more common in the GSRCC population, but this did not impact survival, whereas there were no differences among the two groups in terms of lymph-vascular involvement. However, the latest factor did impact patients’ prognosis at the univariate analysis, both in the N-GSRCC and in the GSRCC patients.

Subdividing patients according to the pathological disease stage (pTNM), we could observe different behaviors in the two groups, as shown in Figure 5. 

The currently used staging system seems to better stratify patients without a GSRC component, even though in both scenarios a statistically significant difference is shown among stages.

These results lead us to evaluate the differences between early gastric cancer (EGC) and advanced gastric cancer (AGC) in the subsets of patients. This is shown in Figure 6.

We showed that in early gastric cancer, the prognosis of GSRCC is better than in N-GSRCC as opposed to advanced cancers (*p* < 0.05). 

The absolute number of harvested nodes did not impact GSRCC patients’ survival, but we showed that both lymph nodal ratio and lymphadenectomy extension have a role in a patient’s prognosis, as shown in Figure 7. 

A D2 lymphadenectomy allows better survival in a statistically significant manner. The median survival of patients undergoing D1, D2, or D3 lymphadenectomy is, respectively, 8, 35, and 16 months.

Finally, we evaluated the postoperative outcome. Recurrence occurred more frequently in patients affected by GSRCC (*p* < 0.05), but in our data, we could not identify a peculiar site of recurrence.

Univariate analysis for the prognostic factors of overall survival for GSRCC was also performed, as stated in the above material and methods section, for other variables. Other statistically significant ones not already shown were preoperative serum albumin level, preoperative N/L ratio, lymph nodes ratio, peritoneal cytology positivity, and completeness of surgical resection. 

In the multivariate analysis, the only independent factors in the prognosis of GSRCC were preoperative albumin level, complete surgical resection, level of lymphadenectomy, and pathological stage. We show this in Table 2.

## 4. Discussion

Based on the GLOBOCAN evaluation, GC serves as the fifth most frequently diagnosed malignancy and the third leading cause of cancer-related death worldwide [12]. 

It is a heterogeneous tumor, indicating different architectural, cytologic, morphological, and molecular profiles. GSRCC is a special variant of adenocarcinoma, and it is defined by the presence of >90% of tumor cells with a large mucin vacuole, which abundantly fills the cytoplasm, resulting in the compression and eccentric displacement of the nucleus, as the recent WHO classification established [4]. 

Unlike the decrease in the incidence of GC seen during the last few years, the research concluded that the incidence of the GSRCC subtype continues to rise [13]. Our study confirmed this epidemiological trend.

To date, many studies are published to better define GSRCC’s clinical impact, but their results are inconclusive. This phenomenon prompts us to re-evaluate this subtype in our surgically treated GC population.

The prognosis of GSRCC and its chemosensitivity to specific chemotherapies are still controversial; it also remains unclear if a specific therapeutic strategy is justified as a benefit of perioperative chemotherapy. The value of taxane-based chemotherapy is unclear, as is whether an up-front surgery is beneficial in locally advanced GSRCC stages.

According to our study, GSRCC patients were younger, equally distributed between males and females, and located almost equally between the proximal and distal parts of the stomach when compared to N-GSRCC ones. Other authors have well described these data [14,15].

Sex impacted survival statistically significantly; females showed better survival in GSRCC patients. It is perhaps because of the well-known estrogenic protective action against tumor aggressiveness added to the fact that GSRCC tumors seemed to induce overexpression of estrogen production in female patients.

As for survival outcomes, there were entirely different long-term survival outcomes for different tumor stages of GSRCC when compared with N-GSRCC. 

We also conducted an analysis of lymph node metastasis of GSRCC and N-GSRCC patients. The results showed that GSRCC patients had a significantly higher incidence of lymph node metastasis than N-GSRCC patients. GSRCC patients also indicated a higher perineural invasion.

A meta-analysis suggested that the frequency of lymph node metastases in early GSRCC is lower than that in non-GSRCC, while there is no significant difference in the frequency of lymph node metastases between advanced GSRCC and non-GSRCC. This suggests that early SRCC may have a different disease evolution compared to an advanced one [16].

Other authors demonstrated this different pattern of lymph node diffusion, such as Jeong SH, who showed that regional lymph node metastases and distant metastasis occur less frequently in mucosal gastric SRCC, but they are associated with an increased risk of cancer-related death when they do happen. Even in the early stages, we should consider surgery as the standard treatment for mucosal gastric GSRCC without considering the endoscopic approach [17].

Based on the difference in lymph node status between early and advanced GSRCC, Zhang et al. compared different lymph node staging systems; the authors concluded that log odds of positive lymph nodes (LOODS) had better predictive prognostic value than pN and lymph node metastasis rate (LNR) both in early and advanced disease. Furthermore, the nomogram constructed in this work by LODDS and clinico-pathological features had good predictive survival performance, indicating that LODDS has good clinical value and is worthy of further application to individuate patients who need intensive follow-up or treatment [18].

Our study showed that signet ring cell carcinoma is not always associated with a poor prognosis; in early gastric cancer, the signet ring cell component is a good and protective prognostic factor. Otherwise, in advanced cancer, the GSRCC subtype seems not to be an independent prognostic factor.

The difference in prognosis between GSRCC and N-GSRCC remains debatable.

Several studies have shown that GSRCC is associated with a worse prognosis than N-GSRC [19,20]. However, Lee et al. [21] reported opposite results in their 2012 paper.

The improved survival shown by our early-stage patients is probably correlated to the younger age of the GSRCC patients at the presentation, and in addition, early-stage GSRCC is associated with less lymph node involvement. Thus, it caused a better prognosis than GSRCC’s advanced stages and N-GSRCC ones. Early-stage GSRCC patients had a surprisingly higher five-year survival rate. These results are consistent with data found in the literature. In addition, Hyung et al. [22] focused their attention on GSRCC early gastric cancer, showing in a cohort of 933 patients a lower involvement of lymph nodes when compared to early non-SRCC cases (5.9% vs. 16.0%, *p* < 0.001). Moreover, he could demonstrate that SRC was an independent protective prognostic factor for node metastasis [23].

Another key point to debate is the role of chemotherapy in GSRCC. To date, there is no strong evidence about the chemotherapy regimen for GSRCC or whether it helps survival outcomes.

In previous studies, GSRCC of the stomach was considered to be non responsive to the common drugs, but there was no clear clinical evidence to support it.

The comparison of chemosensitivity between GSRCC gastric carcinoma and non- GSRCC is still limited in the literature.

An interesting work by Shu Y et al. demonstrated that GSRCC that is not sensitive to chemotherapy is related to the CLDN18-ARHGAP26/6 fusion gene [24].

Phoo NLL et al. [25] found in their research that AKR1C1 and AKR1C3 could play a crucial role in promoting drug resistance by neutralizing the ROS pathway generated by cisplatin. Meanwhile, the inhibition of AKR1C3 and 1C1 effectively up-regulated ROS generation, increasing the cytotoxicity of cisplatin and promoting autophagic cell death while reversing the cisplatin resistance property in signet ring gastric carcinoma patients as a unique biological behavior.

Li, in his well-published work, analyzed the survival of stages II–III primary GSRCC by adjuvant chemoradiotherapy [26]. In this study, he showed that GSRCC patients with stage II–III experienced improved overall survival after receiving adjuvant chemoradiotherapy.

In contrast, we demonstrated that both neo-adjuvant chemotherapy and adjuvant chemotherapy were not beneficial in terms of overall survival in our GSRCC population.

Only aggressive curative surgery seemed to add a survival benefit, in particular if accompanied by an extended and well-made lymphadenectomy.

However, there is still a lack of in-depth understanding of the biological characteristics of GSRCC, and further research is required to formulate targeted treatment strategies for GSRCC. The group of Zhao W. et al. well defined the GSRCC cytological characteristics and its immune microenvironment, which may be advantageous for the accurate diagnosis and treatment of GSRCC [27]. They demonstrated that GSRCC tumors showed a significant presence of the MAPK and estrogen signaling pathways, which could interact and promote each other and continue to amplify each other’s effects. GSRCC cells exhibited lower cell adhesion and higher immune evasion capabilities, as well as an immunosuppressive microenvironment, which could be closely associated with the relatively poor prognosis of GSRCC and the low response to the immunotherapy. In fact, they showed that the sub-clusters of B and T cells in GSRCC have unique infiltration characteristics. The infiltration of follicular B and CD4-T reg cells increased, and that of CD8-T cells decreased in GSRCC tumors, explaining their aggressiveness and infiltrative pattern.

Chen J. et al. [28] studied the GSRCC immune microenvironment (TIME), and they demonstrated that the TIME of advanced GSRCC is enriched for immunosuppressive factors. In addition, they have found that GSRCC TIME showed additional lymphoid structures associated with a high expression level of CXCL13. These findings could be correlated with the bad prognosis and the anti-PD1 treatment resistance of GSRCC. This is an important study that provides an adaptive immune atlas of GSRCC at the single-cell level for the first time, revealing the crucial roles of specific T- and B-cell states in mediating an irresponsive TIME.

## 5. Conclusions

Signet ring cell carcinoma has specific oncogenesis and phenotypes, as well as treatment resistance heterogeneity. Systemic therapies are often ineffective, and predictive biomarkers to guide treatment are urgently needed.

Tumor organoids have recently emerged as an ideal model for drug testing and screening, and we could consider them a useful tool in GSRCC.

According to our multivariate analysis, some prognostic factors could define GSRCC survival, including the preoperative serum albumin level. Nutritional status should be accurately evaluated and balanced, and we should advocate a radical surgical procedure associated with an adequate lymphadenectomy to improve patients’ survival.

This study has some bias as its retrospective analysis was made in a single western surgical center and in the absence of a molecular evaluation of this subtype of gastric cancer.

Ushiku et al. also investigated the role of RHOA mutation in diffuse-type gastric cancer, showing its controversial role; the authors concluded that RHOA seems to be related to diffuse-type gastric cancer but has a limited prognostic impact in isolation [29].

Natsume H. et al. [1] evaluated the mutation spectrum of *TP53* in gastric cancer, demonstrating that it is more frequent in the diffuse subtype of GC. In addition, the divergence in the mutation spectrum of *TP53* in different areas of the world may reflect various pathogeneses and etiologies of GC from region to region. The diversified mutation spectrum of *TP53* in GC may also suggest the different behavior and carcinogenic pathways in the East and West.

Concluding, to respect the latest WHO classification [5], all patients should be reclassified into three groups according to the proportion of cells with signet ring features, as Roviello et al. have just carried out in their work [30]. In this way, we could highlight whether signet ring cell percentage relates to tumor aggressiveness and confirm the role of the GSRCC pattern as a predictor of survival.

## Figures and Tables

**Figure 1 jpm-13-01157-f001:**
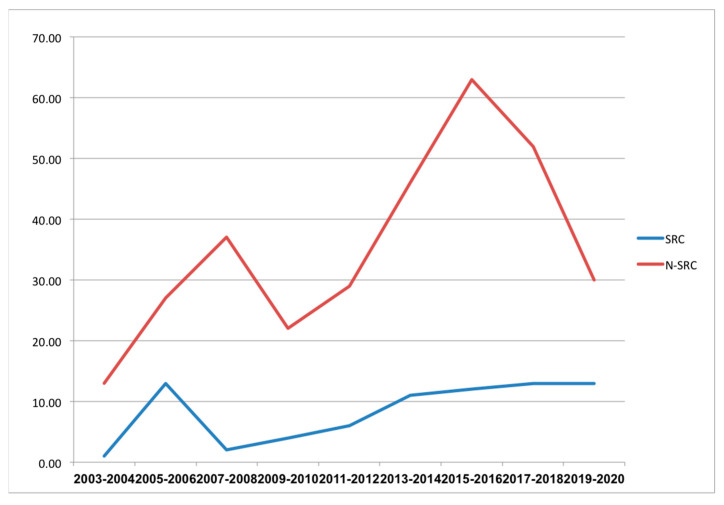
Epidemiological changes in incidence over time; GSRCC incidence is indicating a constant increase significantly different from the N-GSRCC population.

**Figure 2 jpm-13-01157-f002:**
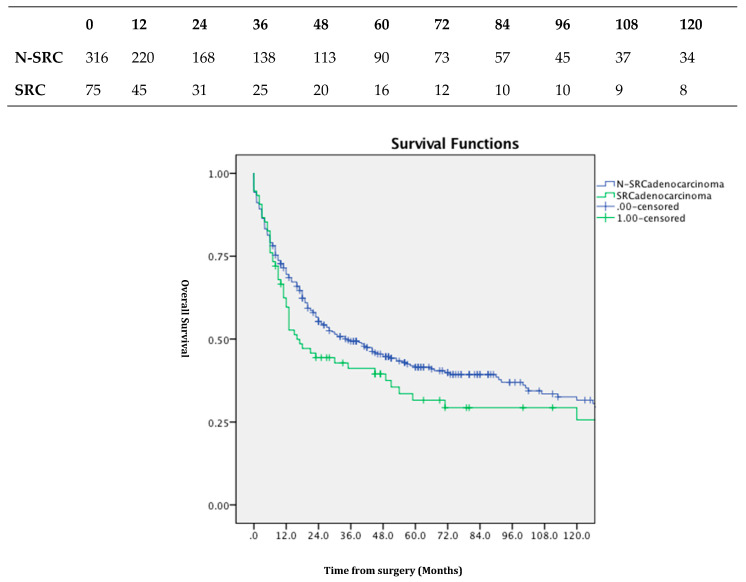
Overall survival according to histological subtypes; SRC and N-SRC do not reach a statistically significant difference; number at risk table.

**Figure 3 jpm-13-01157-f003:**
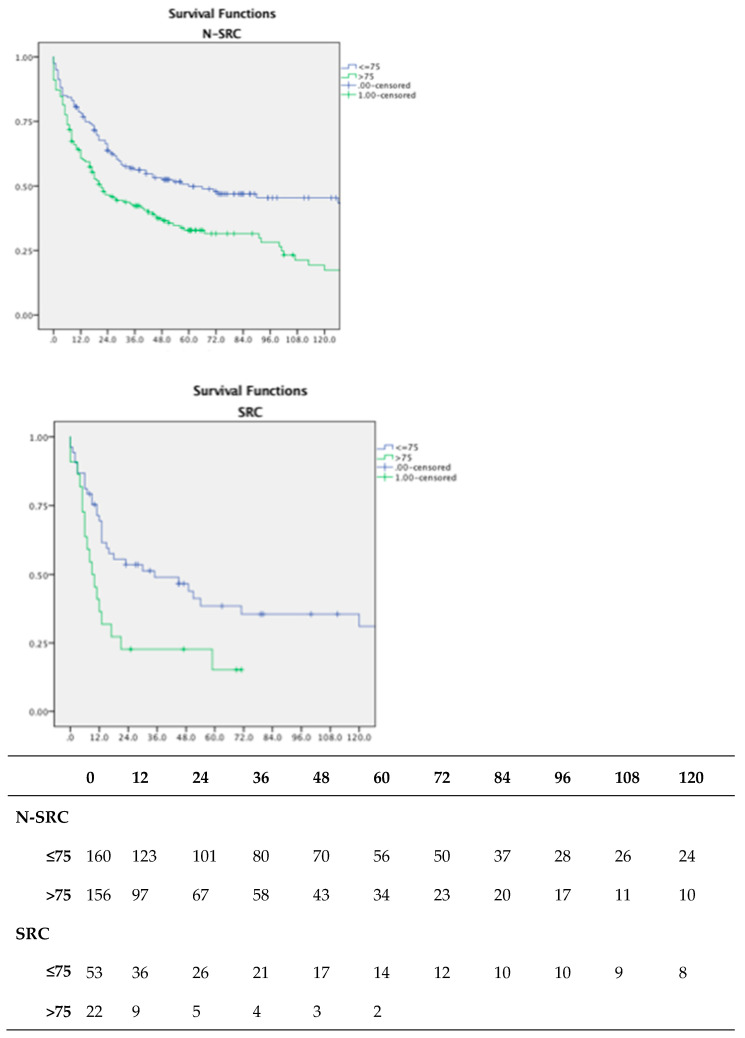
Overall survival according to median age in both N-SRC and SRC cancer patients; number at risk table.

**Figure 4 jpm-13-01157-f004:**
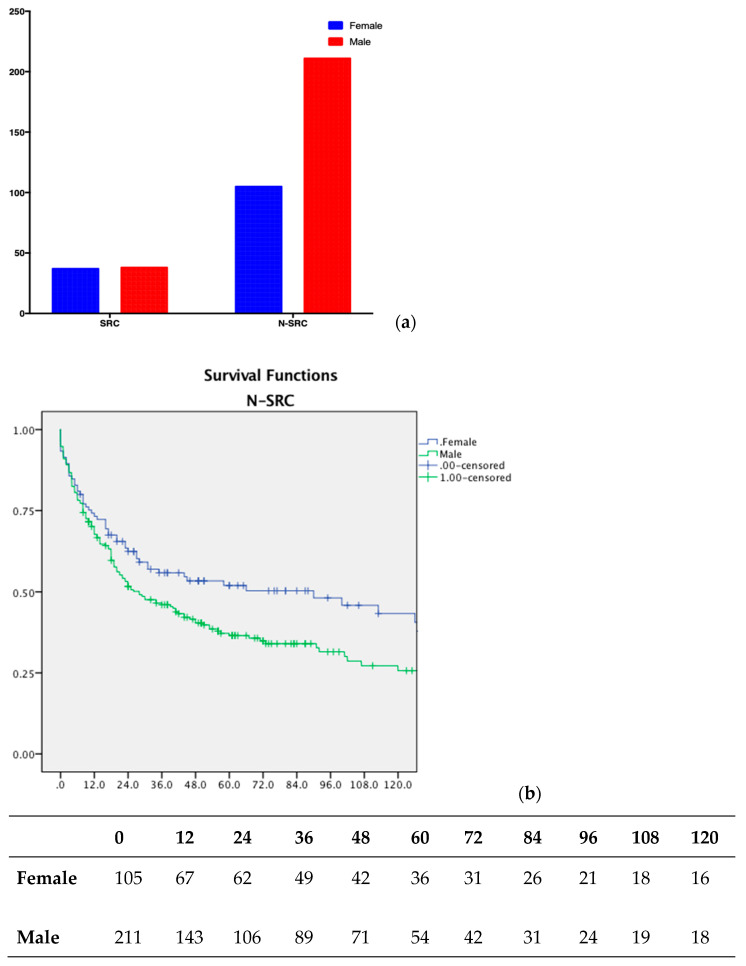
(**a**) Gender distribution among GSRCC and N-GSRCC patients; (**b**) overall survival according to gender in the N-GSRCC group, *p* < 0.05; number at risk table.

**Figure 5 jpm-13-01157-f005:**
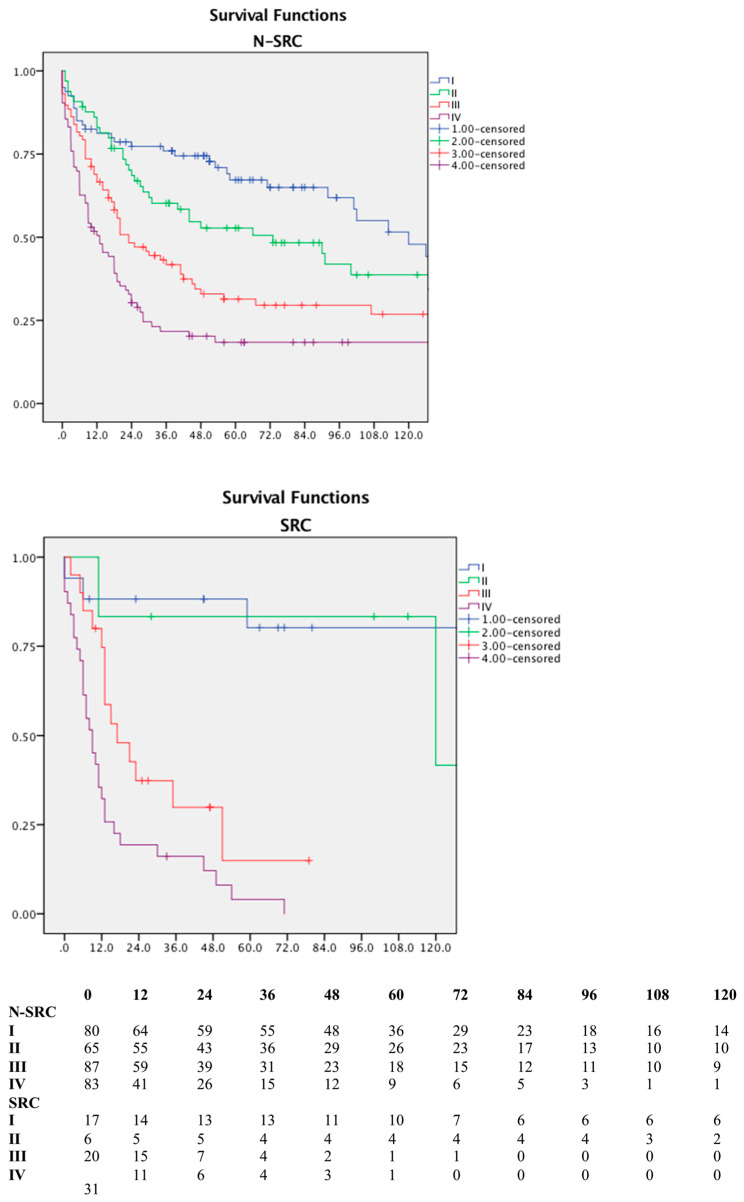
Overall survival according to AJCC/TNM pathological stage; N-SRC and SRC show a completely different stratification of stages; number at risk table.

**Figure 6 jpm-13-01157-f006:**
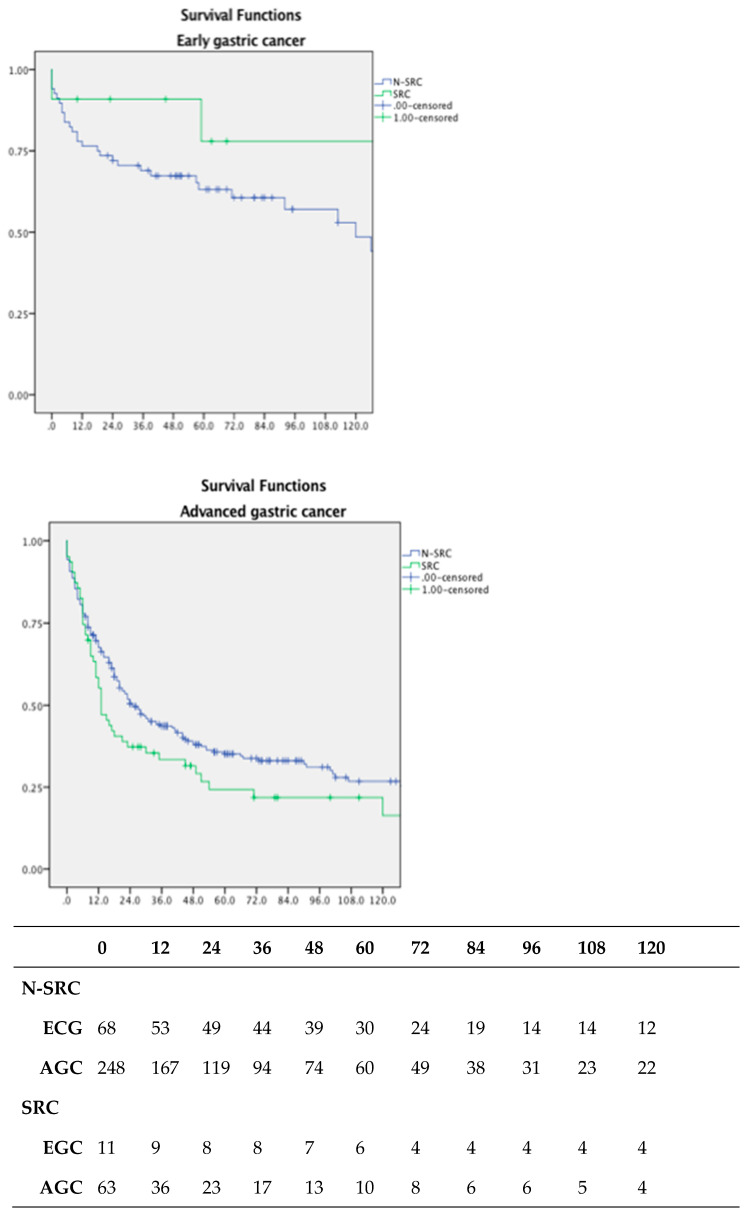
OS according to disease stage; early gastric cancer (EGC) is showing a significantly better survival in patients showing SRC features (*p* < 0.05); this difference is not reached in the advanced gastric cancer (AGC) group and number at risk table.

**Figure 7 jpm-13-01157-f007:**
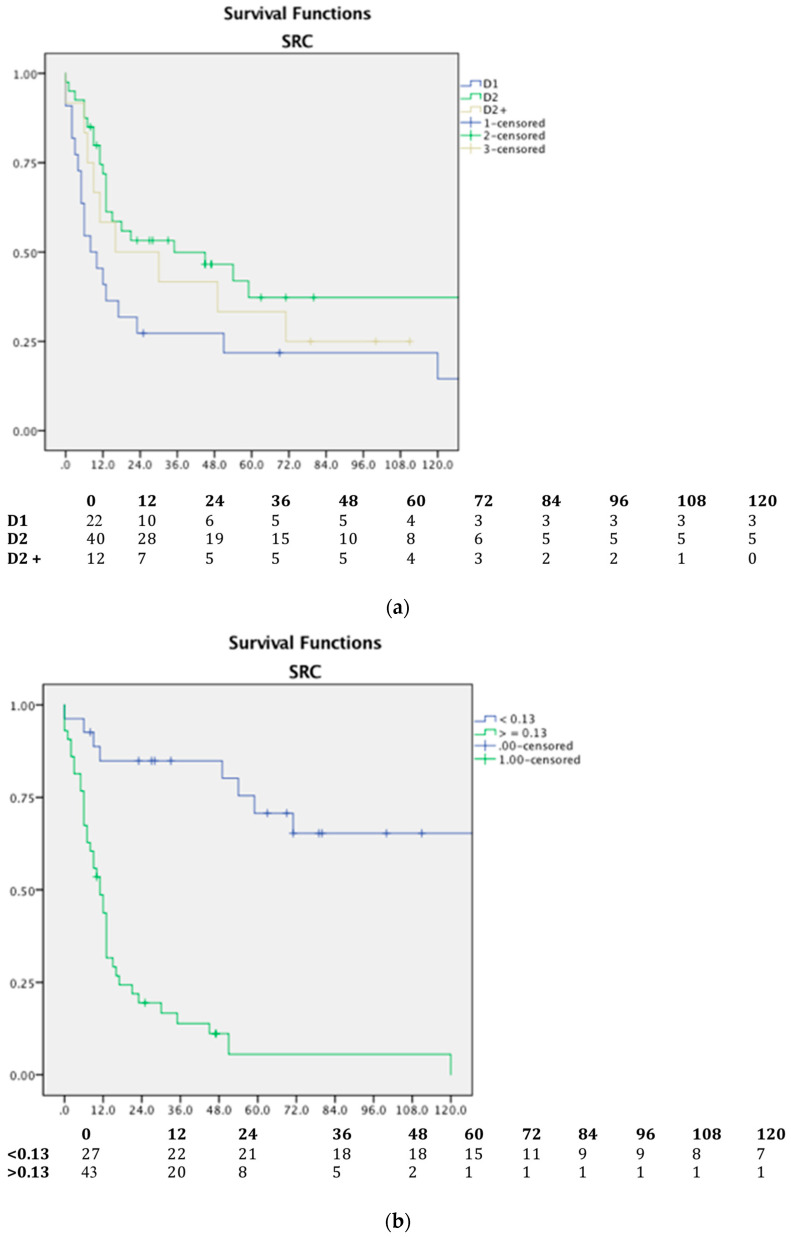
Overall survival in SRC patients according to (**a**) lymphadenectomy extension and (**b**) lymph node ratio; *p* < 0.05 and number at risk tables.

**Table 1 jpm-13-01157-t001:** Patients’ clinic pathological features, among the two groups, were analyzed; *p* < 0.05 is considered statistically significant (N/L: neutrophil to Lymphocyte ratio; EGC: early gastric cancer; AGC: advanced gastric cancer; Ln Ratio: lymph node ratio; Y: yes; N: No).

Clinical Pathological Characteristics			GSRCC (*n* = 75)	N-SRCC (*n* = 316)	*p*
**Age**					
	**<75** **>75**		53 22	160 156	**0.001**
**Gender:**					
	**Male** **Female**		38 37	211 105	**0.004**
**Preoperative Albumin:**					
	**<3.7** **>3.7**		40 33	127 176	0.60
**N/L**					
	**<2.67** **>2.67**		47 25	145 158	**0.007**
**Tumor Location:**					
	**Proximal** **Distant**		31 40	101 199	0.18
**Surgery**					
	**Subtotal Gastrectomy** **Total Gastrectomy** **Other**		40 29 6	199 101 15	0.21
**Lymphadenectomy Level:**					
	**D1** **D2** **D2+**		22 40 12	102 159 46	0.84
**EGC** **AGC**			11 63	68 248	0.26
**pT**					
	**1** **2** **3** **4a** **4b**		11 9 23 21 8	67 39 90 89 25	0.74
**pN**					
	**0** **1** **2** **3a** **3b**		19 5 7 17 24	116 42 59 58 36	0.0001
**Stage**					
	**I** **II** **III** **IV**		17 6 20 31	80 65 87 83	**0.01**
**Lymph node Harvested:**					
	**<27** **>27**		44 26	135 173	**0.004**
**Ln ratio:**					
	**<0.13** **>0.13**		27 43	165 142	**0.02**
**Veno-Lympathic. Invasion**					
	**Yes** **No**		39 11	143 64	0.23
**Perineural Invasion**					
	**Yes** **No**		39 9	114 83	**0.002**
**Peritoneal cytology:**					
	**Positive** **Negative**		17 37	41 166	0.06
**Completeness of Surgery:**					
	**R0** **R1** **R2**		39 23 3	225 45 4	**0.005**
**Chemotherapy:**					
	**Neoadjuvant:**				
		**Y** **N**	10 62	47 265	0.80
	**Adjuvant:**				
		**Y** **N**	19 12	77 52	0.80
**Recurrences:**					
	**None** **Local** **Hematological** **Lymph node** **Peritoneal** **More than 1 location**		36 4 1 2 7 0	190 6 15 2 19 8	0.04

**Table 2 jpm-13-01157-t002:** Multivariate analysis according to Cox regression and Hazard ratio; *p* < 0.05 is considered statistically significant.

Clinical Pathological Variables	Hazard Ratio	95.0% CI		*p*
		Lower	Upper	
**Age**	1.563	0.962	2.540	0.071
**Sex**	1.442	0.898	2.317	0.130
**Albumin Level**	1.629	1.000	2.653	0.050
**N/L Ratio**	0.755	0.494	1.155	0.195
**Venolymphatic. Invasion**	0.909	0.421	1.962	0.807
**Perineural Invasion**	0.805	0.465	1.396	0.441
**Harvested Node**	0.701	0.392	1.253	0.230
**Lymph Nodal Ratio**	1.915	0.890	4.118	0.096
**Surgical Radicality**	2.324	1.421	3.800	0.001
**Type of surgery**	1.027	0.707	1.492	0.887
**Lymphadenectomy**	0.403	0.269	0.603	0.000
**Peritoneal Cytology**	1.013	0.705	1.457	0.943
**pT**	0.928	0.672	1.282	0.650
**pN**	0.975	0.735	1.293	0.860
**pStage**	1.949	1.316	2.885	0.001

## Data Availability

The datasets used and/or analyzed during the current study are available from the corresponding author on reasonable request.

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
