# Peer review of "Prognostic Survival Significance of Signet Ring Cell (SRC) Gastric Cancer: Retrospective Analysis from a Single Western Center"

_jpm, 2023, doi:10.3390/jpm13071157_

Round 1

Reviewer 1 Report

This is an interesting question, but does not fit the scope of a personalized medicine journal without molecular data.

Would recommend going back to the population under study, review and report on MMR data, HER-2 data, PD-L1 data and any other relevant NGS data available for those patients and compare to historical control clinical outcomes.

Would recommend reviewing specific chemotherapy regimens administered, as well as chemo radiation in this studied population, and compare with historical controls. 

As this was a retrospective review, please state the number of patients the analysis was based on in the abstract. 

Overall good English. Clearly written.

Author Response

I strongly agreed with the review and I would like to thank him for the interesting comments. I believe that our paper could fit the scope of "Personalized Medicine Journal" although there isn’t a molecular analysis because the paper focused its attention on a particular subgroup of patients within the entire population. In addition I note that not all the published papers by this journal report molecular characterization.

  • GSRCC is commonly considered to be more aggressive with a greater probability of lymph node and peritoneal metastasis than other gastric cancer types and for this reason it’s characterized by a worse prognosis. Chemotherapy remains the major treatment for advanced GC but GSRCC shows a minor sensitivity to chemotherapic agents, although their recent improvements, compared to non-GSRCC. It is still unclear whether GSRCC patients could benefit from a target therapy and personalized treatment as the few studies existing in the literature demonstrated. Our series is a retrospective and ancient one, including patients from 2004 to 2020. Our pathologists haven’t done in a routinarely way molecular analysis and we recently started to evaluated MSI status, HER-2 and PD-1 in all specimen. While FDA and regulatory agencies in Asian countries (Japan, Korea, and China) have been approved for a long time the combination of nivolumab and chemotherapy irrespective of PD-L1 CPS for advanced not operable GC, in our country guidelines have approved or recommended it for AGC with CPS ≥ 5 since two years. This explains the lack of molecular characterization in the entire population of GSRCC patients. Her-2 was tested only in metastatic not resectable patients. I want to underline that the aim of our study is only to analyse GSRCC survival outcomes compared to non-GSRCC ones and our next objective will be the correlation between molecular features and survival prognosis.
  • Concerning the chemotherapic regimens, I would emphasize the fact that patients underwent to different schemes coming from both different areas of our region and oncological centers. Only neoadjuvant chemotherapy (NAC) was standardized; at the beginning of the NAC we utilized ECF or ECX scheme as Magic trial described; from 2019 following the 2/3 randomized trial written by al Batran, FLOT became the standard of care.
  • We add in the abstract the number of patients.

Reviewer 2 Report

The data itself is not new for the pathologists in endemic area, but the information from Europe (Italy and Eastern Europe) is important the authors should cite the current geopathological data more (PMID: 36600315 ).

Histological subtyping is notoriously subjective, then clarify the following information for international readers:

1. Numbers of blocks to be pathologically investigated to subtype. And were the pathologists took the blocks ? (from the surface or from the deepest front?)

2. Subtyping was usually based on the majority or the greatest area in the cut surface of the whole tumor in Japanese classification system, but how did the authors assign the tumors with signet ring cell carcinoma in part (these mixed one is worse in prognosis).

3. Just exclude the HDGC like clinical phenotype for example multiple sig.

4. I am curious early detection of sig were done in European medical system. Were the subjects symptomatic? or routine check-up disclosed the lesion?

 5. Molecular nature of this entity has been widely known ( PMC4824805) than the author listed.

Author Response

  • The pathologist usually made a number of blocks variables from 3 to 10 . The blocks were taken from the entire layer of the gastric wall and in the past he considered “GSRCC” when the population of signet cell ring tumor is more than 90%. Nowdays we adopte the new WHO classification and we are reviewing and re-classifying all the specimen.
  • No Hereditary Diffuse Gastric Cancer (HDGC) patients belonged to this series.
  • Diagnosis of GC was made mainly for symptomatic patients; in Italy, as in many other European countries, no screening system is available for this disease although our region is a high incident one. This is the reason why the 70% of patients was affected by an advanced GC.
  • Many papers reporting molecular analysis are reported in the conclusion

Round 2

Reviewer 1 Report

Thank your for reply. Would recommend further molecular evaluation of this population and resubmission.

Author Response

I appreciate your comment and suggestion. As we reported in the paper, some authors have evaluated the molecular profile of GSRCC without reaching an univocal conclusion. I would re-underline that the aim of this paper is to only make a picture of Signet ring cell GC population evaluating exclusively the clinical and pathological features. The molecular analysis will be the next future step, infact our pathologist is re-examining specimen focusing attention to MSI staus, MMR changes, Her2 and PD-1 expression comparing the various subgroups according to the WHO classification.

Reviewer 2 Report

It is significantly improved. The practice of pathological diagnosis, especially sub typing like this, is sometimes different among countries. Just obtain the information from the pathologists in. the authors' institute on how many blocks are usually investigated for sub-typing, because in some Asian countries make almost all he cut surface of the resected gastric tumor into blocks and microscopical investigation, which may influence the sampling bias on subclassifications.

Author Response

I strongly agree. Our pathologist routinely increases the number of blocks dependent on histological subtype reaching the number of 9 in poorly cohesive pattern taking up the entire surface of the gastric wall.